# Trading patients' choice in providers for quality of maternity care? A discrete choice experiment amongst pregnant women

Mattijs S. Lambooij[1]*, Jorien Veldwijk[2], Paul F. van Gils[1], Anita W. M. Suijkerbuijk[1], Jeroen N. Struijs[1,3]

**1** Centre of Food, National Institute for Public Health and the Environment, Prevention and Health care (VPZ), Bilthoven, the Netherlands, **2** Erasmus Choice Modelling Center (ECMC), Erasmus School of Health Policy & Management (ESHPM), Erasmus University Rotterdam, Rotterdam, the Netherlands, **3** Department of Public Health and Primary Care, Leiden University Medical Center Campus The Hague, Leiden, the Netherlands

* mattijs.lambooij@rivm.nl

## Abstract

### Background

The introduction of bundled payment for maternity care, aimed at improving the quality of maternity care, may affect pregnant women's choice in providers of maternity care. This paper describes a Dutch study which examined pregnant women's preferences when choosing a maternity care provider. The study focused on factors that enhance the quality of maternity care versus (restricted) provider choice.

### Methods

A discrete choice experiment was conducted amongst 611 pregnant women living in the Netherlands using an online questionnaire. The data were analysed with Latent Class Analyses. The outcome measure consisted of stated preferences in the discrete choice experiment. Included factors were: information exchange by care providers through electronic medical records, information provided by midwife, information provided by friends, freedom to choose maternity care provider and travel distance.

### Results

Four different preference structures were found. In two of those structures, respondents found aspects of the maternity care related to quality of care more important than being able to choose a provider (provider choice). In the two other preference structures, respondents found provider choice more important than aspects related to quality of maternity care.

### Conclusions

In a country with presumed high-quality maternity care like the Netherlands, about half of pregnant women prefer being able to choose their maternity care provider over organisational factors that might imply better quality of care. A comparable amount of women find

**Data Availability Statement:** All relevant data are within the paper and its Supporting Information files.

**Funding:** The current study was funded by the Dutch Ministry of Health, Welfare and Sport as part of a national monitor 'bundled payment for maternity care' by the National Institute of Public Health and the Environment. The Dutch ministry of Health Welfare and Sport was not involved in the design of the study, its administration, the analysis of the results, in the manuscript preparation or submission. The views expressed are those of the authors and do not reflect those of the Ministry of Health, Welfare and Sport or their staff.

**Competing interests:** The authors have declared that no competing interests exist.

quality-related aspects most important when choosing a maternity care provider and are willing to accept limitations in their choice of provider. These insights are relevant for policy makers in order to be able to design a bundled payment model which justify the preferences of all pregnant women.

## Introduction

In many countries, payment reforms are seen as a major lever to achieve new models of care delivery including improved care coordination and multidisciplinary collaboration amongst providers [1–3]. An episode-based approach to payment, known as bundled payment, has been the subject of widespread experimentation [4–6]. Bundled payment provides a single payment for all services by providers for a given condition or treatment [7]. Bundled payment models are increasingly popular in multiple countries, thus enabling countries to learn from successes and failures in the different national systems.

The rationale behind payment models such as bundled payment is that they incentivize care coordination, and consequently stimulate the use of high-quality care by increased provider accountability. Empirical associations have been found between use of integrated care and increased patient satisfaction, perceived quality of care and patient access to services [8]. Use of a bundled payment is a shift from the commonly used fee-for-service models in which the financial risks are primarily borne by the payers. A fee-for-service system supposedly enables providers to deliver larger volumes of care regardless of the quality of their services [3, 9–12]. Bundled payment models align financial incentives with desired health outcomes and spending and by doing so aim to motivate providers to deliver high quality of care [3]. Bundled payment serves as an incentive to avoid unnecessary care, and encourages more efficient coordination between care practitioners. In the Netherlands, bundled payment for diabetes care had been introduced in 2007 and had both positive and negative consequences. Positive consequences included quality improvements in care delivery processes and in the transparency of delivered care. Negative consequences included antitrust concerns and more limited choice for patients [13].

In 2017 in the Netherlands, pilots regarding bundled payments for maternity care have been implemented. In six regions, a provider-led collaborative group, called an Integrated Maternity Care Organization (IMCO), is the general contractor for the period in which maternal care is provided and subcontracts care providers to deliver the actual care. More background information on the Dutch health care system, specifically the maternity care system and the bundled payment model for maternity care can be found in Supporting information S1 File.

Currently, the question arises whether the bundled payment for maternity care limits pregnant women's choice of providers. Potentially, subcontracted providers of an IMCO might steer pregnant women toward subcontracted providers of the same IMCO. This potential limitation in provider choice already occurs at the beginning of women's pregnancies as their first contact with a midwife or gynaecologist implicitly determine the hospital in which women will deliver their children as in each IMCO only one hospital participates. To guarantee unlimited provider choice, the Dutch bundled payment model includes a a so called "bundle breaker" which in practice means that if women receive service delivery from care providers whom were not subcontracted by the IMCO they are free to do so. Also the services of not subcontracted providers are reimbursed. In order to support the debate regarding the design of the

bundled payment model—especially the desirability of the bundle breaker—more insights in the preferences of pregnant women are essential.

In the considerable body of evidence on preferences of pregnant women on maternity care, aspects related to provider choice [8–14] and aspects of maternity care related to quality of care [11–13, 15–18] are identified to be relevant. But also personal experiences [12, 14, 15, 19–21], birth place location [13, 21–23], and influence in the birth process [14] have been found to affect the preference of pregnant women. Pain relief is found to be important by both the women and their partners and in turn this is influenced by previous experiences [24]. Also previous DCEs have found that women value nurses appointments during being pregnant [18] and that women are willing to accept fewer visits by maternity care professionals, if this is replaced with sufficient alternative care [17]. S2 File provides a table with all factors affecting pregnant women's preferences which the study identified. In this study we focussed on aspects that are indications of or affect quality of care and provider choice. The results are meant to support the policy debate. In this paper we studied pregnant women's preferences regarding maternity care services, specifically the relative importance of their preferences concerning quality of care and choice in provider. We used the preference structures that resulted from the DCE to draw conclusions about the extent to which women's personal preferences concerning quality of maternity care and choice in provider are sufficiently met under a bundled payment regime.

The results of this study are relevant for policymakers, as they provide information on the preferences of pregnant women with respect to choosing their maternity care providers. The results will show whether it is desirable to maintain the pregnant women's unlimited choice in providers via the bundled breaker, or whether women are willing to accept (some) limitations in the list of providers they may be able to choose from. The study provides information on the willingness of pregnant women to trade between patients' choice in providers and quality of maternity care aspects, within the context of the Dutch experiment with bundled payments and related IMCOs.

## Materials and methods

We used a Discrete Choice Experiment (DCE) to study the preferences of pregnant women when they choose maternity care providers. Subsequently, we analysed how pregnant women in the Netherlands weigh unlimited provider choice relative to better higher quality of care.

### DCE

Discrete Choice Experiment (DCE) is a stated preference method used to elicit individuals' preferences by quantifying the relative importance of the characteristics of, for instance, health care services [15, 16]. In the DCE we asked pregnant women to choose between two hypothetical maternity care providers. Respondents were asked to complete nine of these 'choice tasks'. Each choice task consisted of two alternative maternity care organisations. In the current DCE, the description of the maternity care organisation was based on its characteristics or 'attributes'. Similar to DCE studies in general, it was assumed that the respondents' preference for an alternative was based on the levels of the included attributes. We analysed the choices that the respondents made thus quantifying their preferences [17].

### Selection of the attributes and levels

In this study, a literature review was conducted to compile an initial list of potential attributes of pregnant women's preferences in health care services (S2 File for complete list potential attributes found in the search). To include recent scientific insights about pregnant women's

preferences, the search was performed on publications from 2010 onwards in PubMed, written in English. The search for Dutch papers yielded no results. The following keywords were used: pregnancy, maternity care, perinatal care, preferences, discrete choice experiments, and DCE. This list was thereafter discussed with four experts (i.e. patient organization representative, research organization specialised in health services research, a national governmental remuneration organization and a director of a maternity care organization) to select relevant attributes considering current practice, as well as to assign levels that respondents can relate to and are close to policy practices.

After the literature review, a pilot study was conducted based on think-aloud interviews, to further assess the relevancy of the attributes, as well as to test and refine in the attribute (level) selection to suit the target population. In total, ten pregnant women completed a pilot questionnaire while also discussing their opinions. In a midwifery practice we randomly asked ten pregnant women to fill out the questionnaire and give feedback. The midwifery practice was located in the center of a medium-sized city. This led to the final set of attributes (rephrasing) and levels (Table 1). The first three attributes are operationalisations of quality of care that can be observed by clients (indicated by a (Q) in Table 1); the other two attributes are operationalisations that affect women's choice in provider. The attributes and levels were introduced and explained to the respondents before presenting the choice sets (S3 File for introductory text).

This process led to the final set of attributes, levels and final rephrasing of attributes and levels (Table 1). The first three attributes are operationalisations of quality of care that can be observed by clients (indicated by a (Q) in de table), the final two attributes in Table 1 are operationalisations that affect the choice possibilities (C) of the women. The first attribute, "Information between care professionals by single EMR" is included to operationalize an essential part of integrating care services: access to relevant and up-to-date patient information by all care professionals [18]. Both the experts and the respondents in the pilots agreed to the relevance of this attribute. The second attribute, 'Information to you by the midwife' was included

**Table 1. Attributes and levels DCE maternity care.**

| | Level 1 | Level 2 | Level 3 |
|---|---|---|---|
| **Information between care professionals, single EMR† yes/no (Q)** | Your health care providers exchange information by email, phone and fax. Consequently, there is a chance that they do not have all the information at hand | All your health care providers work from a single patient file and are therefore always well informed about your situation | |
| **Information to you by the midwife (Q)** | Information about pregnancy and delivery, able to ask questions via phone calls, email, texting. | Information about pregnancy and delivery, able to ask questions during first meeting, leaflets | Information about pregnancy and delivery, able to ask questions, request information via leaflets, websites, questions can be asked during later stages of pregnancy |
| **Advice by friends and family (Q)** | You hear positive stories about this maternity care organisation from your friends and family (word of mouth) | You hear both positive and negative stories about this maternity care organisation from your friends and family (word of mouth) | |
| **Organisation of maternity care (C)** | All care organised separately | Usually, all care delivered by one organization, but also, free choice outside of the organization.* | All care obligatory in one organisation (no choice in provider)** |
| **Travel time(C)** | The midwife practice is located at 5 minutes cycling distance from your house; the hospital is a 5 minute drive | The midwife practice is located at 10 minutes cycling distance from your house; the hospital is 10 minute drive | The midwife practice is located at 15–20*** minutes cycling distance from your house; the hospital is a 15–20*** minute drive |

* In Table 2: Organization part fixed, part choice

**In Table 2: Organization no choice

***Two different levels: 3 and 4 contained 15 and 20 minutes respectively for both midwife and hospital.

† EMR = Electronic Medical Record

to operationalise the way the information from the health care organisation was given to the pregnant women. In order to make informed decisions, having access to sufficient and accurate information is essential. The third attribute 'Advice by friends and family' was an operationalisation of quality of care of the maternity care organisation. The Dutch National Health Care Institute (Zorginstituut Nederland) developed a set of quality indicators for maternity care [19]. Following this set, in expert interviews and in the pilot test we tested the following options: (1) a standardized quality score on internet indicating whether the organisation was ranked in the top 20% or average, (2) percentage of complications of the maternity care organisation, presented on the internet, (3) a score on forced deliveries, presented as average, below average or above average, (4) average APGAR score of maternity care organisation, (5) reviews on the internet of experiences by other women. The experts preferred more the objective measures (1–4), however, the respondents in our pilot study indicated that this information did not mean anything to them and that they would only value information related to quality of care presented to them by people that they knew. The fourth attribute 'Organisation of maternity care' was included to operationalize the consequences for the pregnant women of an integrated care organisation (IMCO). The final attribute 'travel time' was included as a consequence of integrating care; if maternity care organisations would integrate, this could imply that the nearest provider would not be cooperating with the IMCO that was chosen but women would need to travel for a longer time for their maternity care visits.

## Experimental design of DCE and survey format

Ngene 1.0 software was used to create a Bayesian D-efficient design with a total of 24 unique choice sets divided over three blocks [20, 21]. Each choice set consisted of two alternatives. Before respondents answered the choice sets, they were offered explanation of the attributes, levels as well as how to complete a choice set, followed by an example. Respondents were asked to choose their preferred maternity care organisation, and whether they would accept the organisation of choice in real life or not (i.e. opt-out) [22]. The questionnaire further contained questions about demographics, gravidity status, and experience with complications during pregnancy.

To estimate the required sample size, we used a generally accepted rule of thumb:

$$\text{Sample size} > 500 \, l/ \, TA$$

Where T = the number of choice tasks, A = the number of alternatives in a choice set, and l = largest number of levels in any attribute (l). In our case a DCE needs at least 84 respondents to estimate the main effects. Given proposed latent class analysis (including heterogeneity) and blocked design (asking people to only complete a subset of the choice tasks, thereby limiting the burden on every single participant) we estimated to need 504 (84*3blocks*2(model adjustment)) respondents.

## Recruitment

We used the database of a panel that covers 75% of all women who ever gave birth and/or who are pregnant in the Netherlands. Pregnant women in the Netherlands can sign up to receive "The happy box", a box that contains number of items such as a cream, a pacifier and clothing, that are useful when caring for a new-born baby. The happy box is organised by a large number of collaborative stores. Many women apply for this box and are subsequently included in this database. For this study, all registered women who were pregnant for more than 12 weeks at the time of data collection were approached by email. The email contained an introductory text to explain the aim of the study and a link to the questionnaire. All women were emailed

once, no reminders were sent. Women were able to stop filling in the questionnaire at any time. Only the questionnaires that were filled in completely were included in the analyses.

### Ethics approval

The study protocol was sent to The Medical Research Ethics Committees United (MEC-U). The MEC-U found the Law Medical Scientific Research not applicable to this study (registration number W15.093).

### Analyses

To analyse the choices of the respondents, Latent Class Analysis (LCA) was conducted. By means of LCA, preferences across unobserved subgroups of the population can be examined [23–26]. Class membership is latent, meaning that each respondent has a certain probability to belong to a class.

The systematic utility component (V in equation 1) describes the observable utility that respondent 'r' belonging to class 'c' reported for alternative 'a' in choice task 't'. The $\beta_0$ represents the alternative specific constant for the opt-out and $\beta1 - \beta_7$ are the attribute level estimates that represent the relative importance of each attribute level. A significant attribute estimate within a certain class indicates that this attribute contributes to the decision of respondents in that class.

**Equation 1: Utility function.** $V_{rta|c} = \beta0_{|c} + \beta1_{|c}$ Single EMR $._{rta|c} + \beta2_{|c}$ Info midwife med$_{rta|c} + \beta3_{|c}$ Info midwife max$_{rta|c} + \beta4_{|c}$ advice friends pos$_{rta|c} + \beta5_{|c}$ Org.partly fixed$_{rta|c} + \beta6_{|c}$ Org.no choice$_{rta|c} + \beta7_{|c}$ travel time$_{rta|c}$

All variables were effect coded [27], except for travel time. This was recoded as follows: 5 minutes = 0,5; 10 minutes = 1; 15 minutes = 1,5 and 20 minutes = 2. The reference category for 'info midwife med' and 'info midwife max' was 'info midwife minimum' (level 1 in Table 1). The reference category for 'Org. partly fixed' and 'Org. no choice' was 'all care organized separately' (level 1 in Table 1).

A class assignment model was fitted to test which respondent-level characteristics can explain class assignment of respondents. We tested the following parameters for a significant contribution to the class assignment model: education high/low, urbanisation level (5 categories), urbanisation city/rural, first child/gravidity status.

The final class assignment utility function was:

$$V_{rc} = \beta0_{|c} + \beta1_{|c} \text{ Urbanisation level home}_r + \beta2_{|c} \text{ high educational level}_r$$

The largest difference value received an importance score of one, representing the attribute that was deemed most important by respondents, the other difference values were divided by the largest difference value, resulting in a Relative Importance Score (RIS) of all attributes to the most important attribute. This means that the attribute with a RIS of 1 is the most important attribute in that particular class, and the sizes of the other scores are indicative of their relative importance compared to the attribute with RIS of 1. Based on model fit (2 log likelihood) improvement, we determined the optimum in number of classes.

## Results

### Sample

A sample of 35,016 pregnant women received the online questionnaire. Of these women, a third (n = 12,956: view rate: 37.0%) [28] opened the e-mail. Subsequently, 2,073 women started to fill in the questionnaire (participation rate: 16%), and about a third of them completed the

questionnaire (completion rate: 30%). No reminders were sent. This yielded in a study population of 611 women.

The mean age of the study population was similar to the mean age of all pregnant women in the Netherlands included in the national monitor 'bundled payment for maternity care' (30.8 vs. 31.1 years) [29]. The study sample contained more highly educated women than in the general population of pregnant women in 2018 aged 15–45 in the Netherlands [30](23.7% vs 14.1% in tertiary education), fewer women from very strongly urbanized regions (18.8% vs 24.8%) and more women who were pregnant with their first child (primagravida) than in the population of pregnant women Netherlands in 2015 through 2016 (73.0% vs 43.8%).

## Results of latent class analysis

A model with four classes yielded the optimal model fit, based on -2 log likelihood improvement (Table 2).

We found four distinctive preference structures in the data. The Class Assignment Probabilities (CAP) indicate the average probability of the respondents to belong to one of the classes respectively. The largest CAP is 0.29 and the lowest of the four is 0.21 (Table 2), which indicates that the classes are evenly distributed and that respondents have about a one in four chance to belong to one of the classes. Educational level significantly contributed to class assignment, meaning that respondents with lower educational levels had the highest probability to belong to class 2 while respondents with a higher educational level had the highest probability to belong to class 4.

The relative importance score (RIS) of the attributes and levels is presented as the relative contribution to the sum of the mean effect sizes. Per attribute, the difference between the highest and lowest attribute level estimate was calculated. People in class 1 found the assumed quality-enhancing use of a single EMR for all maternity care providers most important (RIS = 1 in Table 2), followed by travel time (RIS = 0.65) and the opportunity to choose one provider in the care organisation (RIS = 0.62). The word of mouth information on quality (RIS = 0.43) and midwife (RIS 0.28) was weighed less important.

For people in class 2, the two most important attributes were related to provider choice: possibility to choose a provider outside of the maternity care organisation (RIS = 1) and travel time (RIS = 0.52). Next, the three aspects related to quality were weighed: working form a single EMR (RIS = 0.51), advice of friends (RIS = 0.17), and the information given by the midwife (RIS = 0.10).

People in class 3 weighed both quality and choice most equally from the four classes. These people found it most important that care providers work from a single EMR, subsequently, the opportunity to choose a care provider outside of the maternity care organisation (RIS = 0.90), then they weighed the information of the midwife (RIS = 0.31) and advice of friends (RIS = 0.26) and finally travel time.

People in class 4 conversely found travel time the most important aspect when choosing for a maternity care organisation, followed by the opportunity to choose an external care provider (RIS = 0.60). The three quality proxies advice of friends (RIS = 0.47), working from single EMR (RIS = 0.45) and information of the midwife (RIS = 0.37) was weighed subsequently. This means that for people in class four, the opportunity to choose was most important.

In summary, we found four reference structures, each with about equal CAPs (Table 2, bottom row). Two of the classes found quality related aspects most important, and two classes found choice related aspects more important. When translating this to actual choices, we expect about half the women to find quality-related aspects to be most important (but also weigh choice-related aspects), and about half to find choice-related aspects most important

**Table 2. Latent class model.**

| | Class 1 | | Class 2 | | Class 3 | | Class 4 | |
|---|---|---|---|---|---|---|---|---|
| | Parameter | RIS[X] | Parameter | RIS | Parameter | RIS | Parameter | RIS |
| | (se) | (Rank) | (se) | (Rank) | (se) | (Rank) | (se) | (Rank) |
| Quality of care aspects | | | | | | | | |
| Single Electronic Medical Record | 1.18 ** | 1.00 (1) | 0.24** | 0.51 (3) | 1.01** | 1.00 (1) | 0.51** | 0.45 (4) |
| | (0.12) | | (0.05) | | (0.08) | | (0.07) | |
| Information midwife medium[a] | 0.36** | 0.28 (5) | 0.06 | 0.10 (5) | 0.32** | 0.31 (3) | 0.44** | 0.37 (5) |
| | (0.10) | | (0.05) | | (0.07) | | (0.06) | |
| Information midwife maximum[a] | -0.06 | | -0.03 | | -0.01 | | -0.03 | |
| | (0.09) | | (0.05) | | (0.06) | | (0.06) | |
| Advice friends | 0.51** | 0.43 (4) | 0.08* | 0.17 (4) | -0.26** | 0.26 (4) | 0.53** | 0.47 (3) |
| | (0.10) | | (0.03) | | (0.05) | | (0.06) | |
| Provider choice aspects | | | | | | | | |
| Organisation part fixed, part choice[b] | 0.89** | 0.62 (3) | 0.52** | 1.00 (1) | 0.85** | 0.90 (2) | 0.77** | 0.60 (2) |
| | (0.10) | | (0.06) | | (0.07) | | (0.06) | |
| Organisation no choice[b] | -0.32** | | -0.10* | | 0.12) | | -0.18** | |
| | (0.12) | | (0.05) | | (0.08) | | (0.06) | |
| Travel time | -0.61** | 0.65 (2) | -0.21** | 0.56 (2) | -0.05 | 0.06 (5) | -0.91** | 1.00 (1) |
| | (0.12) | | (0.06) | | (0.08) | | (0.09) | |
| Constant | 1.64** | | -0.34** | | 0.69** | | -1.21** | |
| *Class predictors* | (0.20) | | (0.30) | | (0.15) | | (0.13) | |
| Degree urbanisation | -0.17 | | -0.49 | | -0.48^ | | | |
| | (0.28) | | (0.30) | | (0.28) | | | |
| Education | -0.74* | | -1.20** | | -0.90** | | | |
| | (0.30) | | (0.32) | | (0.31) | | | |
| Intercept | 1.64** | | -0.34 | | 0.69** | | -1.21** | |
| | (0.20) | | (0.30) | | (0.15) | | (0.13) | |
| Class assignment probabilities | 0.23 | | 0.21 | | 0.29 | | 0.28 | |

** = p<0.01

* = p<0.05

^ = p<0.1

X = Relative important scores: 1 = most important, other values relative importance compared to 1

[a]: reference category = minimum information

[b]: reference category = no choice within organisation

(but also weigh quality aspects) when choosing maternity care organisations. The medium option in the attribute of organisation of care (Organisation part fixed, part choice in Table 2) was valued as more attractive than complete provider choice in all four preference structures.

## Discussion

Our main objective was to gain insight in the willingness of pregnant women to trade between choice in providers and quality of maternity care aspects.

We found four different preference structures amongst pregnant women in choosing maternity care providers. Educational level is the main predictor of these differences. For some women the unlimited provider choice is the most important aspect and they find this more important than the aspects related to quality of care In contrast, other women are willing to accept a limited choice in a maternity care organisation and providers when services are

better coordinated and integrated, resulting in better quality of care. Our results are in line with previous findings that women are willing to trade between factors of maternity care [31]. What this study adds to the literature is the diversification in the population of interest: different women will make different choices when they can. This is in line with the previous finding that women differ in preferences related to service attributes [32]. A difference between that review and our findings is that the current study is restricted to attributes related to organisational aspects of maternity care services that are directly influenced by the design of the bundled payment model, while the underlying studies in the review also included other aspects (e.g. personal experiences, factors related more specifically to maternity care providers involved, such as continuity of midwife) [32]. We excluded personal experiences [33, 34] women's beliefs and values [35] or type of birth setting [36–38], even though these factors are known to be relevant and to affect the preferences of pregnant women. The downside of restricting this study to factors that can be affected by the organisation of care is that we do not know the relative importance of those factors compared to the factors we did not include.

We also found that availability of pain relief was very important to women and partners [14, 32, 36–42]. Although pain relief is generally considered a highly important aspect of maternity care decisions, this is only very weakly linked to the organizations of integrated care and therefore not included in this study.

For the majority of the respondents we found that the information given by the midwife was weighed to be least important, and information given by friends and family was more important. This is in line with [43] who found that midwifes were not the main source of information for pregnant women, but that women wanted the option to discuss and consider their birth preferences throughout their pregnancy with multiple sources, not at a fixed point. This would mean that policymakers nor health care professionals are able to provide pregnant women with the type of information that they need in order to develop their preferences.

This study particularly focussed on factors of maternity care that can be influenced by healthcare providers and policymakers. In this choice experiment, we excluded personal experiences [33, 34] women's beliefs and values [35] or type of birth setting [36–38], even though these factors are known to be relevant to affect the preferences of pregnant women. The downside of restricting this study to factors that can be affected by the organisation of care is that we do not know the relative importance of those factors compared to the factors we did not include. However, the advantage is that this study provides practitioners and policymakers with relevant information on how maternity care clients will respond to changes in how their maternity care is organised. Future studies may yield relevant new information on both relative importance and possible interactions of difficult-to -measure aspects such as ofand harder-to-affect aspects such as women's previous experiences compared to factors that can be affected by maternity care providers or policymakers.

Current design of the Dutch bundled payment guarantees an unlimited choice for pregnant women via the existence of the bundle breaker. In practice, this means that when a pregnant women receives care from a care provider, which is not subcontracted by the IMCO, the bundled payment expires and all maternity care services are reimbursed via the existing payment model which is predominantly fee-for-services. This leads to an enormous administrative burden for care providers. In order to keep support among care providers for the bundled payment model the design of the bundle breaker preferably must be simplified in order to find a middle ground between unlimited provider choice and the administrative burden for care providers. Finding this optimum is not straight forward but is needed to keep support among both pregnant women and providers. The results of this study support current policy design of the bundle breaker, indicating that the current of the bundled payment with a bundle breaker appears to be close to the optimum of preferences of pregnant women.

A number of methodological aspects needs to be kept in mind when interpreting the results.

Firstly, the overall response rate was relatively low. Pregnant women are particularly hard to reach, and are less willing to participate in research [44]. We found that in our sample, highly educated women, women from rural areas and women who were expecting their first child were overrepresented. Previous research has shown that higher educated respondents trade more of the included attributes than lower educated respondents in DCEs [45]. This could mean that in real life choices, a larger proportion of women weigh fewer of the maternity care aspects than the aspects that were included in the current study. Despite the overrepresentation of highly educated respondents, the sample benefits from the fact that all respondents had actual experience in the choice in the DCE because they were pregnant at the time of filling in the questionnaire. This meant that respondents were familiar with and had experience in the decisions they were asked to make in DCEs. In the analyses, we found that level of urbanisation, gravidity status did not predict the classes in the LCA. This implies that these person related factors do not affect the preferences structures that we found. If we would have had a sample with more highly educated women, living in a city it is likely that the parameters we would have found would differ. However, considering the relative importance of the attributes in the four preference structures, these differences in parameters would need to be very large to affect the ranking, and therefore the conclusions that we draw on the existence of the preference structures. We have therefore no reason to assume that the relative importance of the attributes in our results differ from the relative importance in real life choices, when just looking at the ranking of attributes. However the precise parameter values are likely to differ between the sample in this study and the general population [46]. We have therefore no reason to assume that a different sample would lead to very different conclusions concerning the preference structures of pregnant women in the Netherlands.

Secondly, and comparably to all DCEs, the number of attributes (i.e. factors presented in the scenarios) is limited, in this case to five. This is a limitation given that a larger number of factors are likely to be taken into account, as we found in the literature scan. For instance, pain relief was an important factor both in the literature scan and in the pilot study. However, since this is highly correlated with place of birth and this is affected by personal aspects such as complications during pregnancy and giving birth, it fell outside of the scope of the study's objective. Nonetheless, by conducting a pilot study on the questionnaire and involving experts in the development of the scenarios, it is assumed that the five most important attributes are included in the DCE. A distinctive feature of a DCE is that it provides information on the relative importance of the included factors [15].This information is hard to measure with other research instruments, while in real life, similar trade-offs are an intrinsic fact of almost every choice. We therefore believe that for policymakers, the current DCE can be used to understand the most important factors that women find important when choosing maternity care services.

Thirdly, within the scenarios quality of care was operationalized as 'advice from friends'. During testing, we tried a number of more objective measures such as reported number of caesarean sections, an online aggregate score presented by independent organization or average APGAR scores. Some of the interviewed experts preferred those more objective quality indicators, but the pilot's participants (pregnant women) almost all indicated that they would not consider such types of information, but that they would rather take into account their friends' or family's opinions when choosing maternity care providers. This finding is in line with previous studies that found that women's preferences are shaped by their home and family context, beliefs and previous pregnancy experiences [39]. Since DCEs' main objectives are to describe scenarios as close to a real decision making situation as possible [46], we decided to include the advice of friends as a quality indicator instead of more clinical quality information.

Finally, in the current study, we formulated a number of factors to represent the quality of care. However, in a real life situation, a large proportion of pregnant women will not know whether healthcare providers work from the same EMR, or how closely the nearest hospital cooperates with the midwifes or maternity care provider. This means that in the current study, the respondents are likely to have more information on factors that affect the quality of maternity care than in real life. Consequently the relative importance of those factors may be overestimated compared to clearly visible factors such as travel time or advice from the midwife. This implies that even though this study yields insights into the preferences of women using maternal care, in practice the women may not be able to realise their preferences due to lack of information regarding the real-life organisation of healthcare providers.

Measuring quality of care is difficult and depending on the study's aim, different angles can be chosen [e.g. 47, 48, 49]. In our case, we needed to exclude more medical technical quality measures (e.g. APGAR) because our main focus was to resemble a real-life choice as much as possible. When it comes to quality of care, it may be useful to compare the relative importance of medical technical quality measures of care and more patient- centred measures of quality of care in future studies.

## Conclusions

Both quality of care and provider choice aspects are important to pregnant women when choosing their maternity care providers. This study gives providers and policymakers relevant information on how Dutch pregnant women respond to changes in the organisation of maternity care. Our results indicated that a substantial number of clients or patients are willing to accept a limited choice when quality improvements are realized, but also that other women attach greater value to more choice in providers. This study indicates that enabling provider choice under a bundled payment model is essential in order to develop delivery models that in meets the demands of all pregnant women.

## Supporting information

**S1 File. Background of Dutch health care system and bundled payment model for maternity care.**
(DOCX)

**S2 File. Long list potential attributes.**
(DOCX)

**S3 File. Introductory text to DCE questionnaire.**
(DOCX)

**S4 File.**
(CSV)

**S5 File. Codebook to dataset.**
(DOCX)

## Acknowledgments

The authors thank Hester van Dorst, Eline de Vries and Hanneke Drewes for thorough reading and commenting on the manuscript.

## Author Contributions

**Conceptualization:** Mattijs S. Lambooij, Paul F. van Gils, Anita W. M. Suijkerbuijk, Jeroen N. Struijs.

**Data curation:** Mattijs S. Lambooij.

**Formal analysis:** Mattijs S. Lambooij, Jorien Veldwijk.

**Funding acquisition:** Jeroen N. Struijs.

**Investigation:** Mattijs S. Lambooij, Paul F. van Gils, Anita W. M. Suijkerbuijk.

**Methodology:** Mattijs S. Lambooij, Jorien Veldwijk, Jeroen N. Struijs.

**Project administration:** Mattijs S. Lambooij.

**Supervision:** Mattijs S. Lambooij, Jeroen N. Struijs.

**Validation:** Mattijs S. Lambooij.

**Writing – original draft:** Mattijs S. Lambooij, Jeroen N. Struijs.

**Writing – review & editing:** Mattijs S. Lambooij, Paul F. van Gils, Anita W. M. Suijkerbuijk.

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
