## [Decision Letter · Decision Letter 0]

25 Sep 2019

PONE-D-19-23167

Trading freedom of choice for quality of maternity care? A discrete choice experiment amongst pregnant women

PLOS ONE

Dear Dr. Lambooij,

Thank you for submitting your manuscript to PLOS ONE. After careful consideration, we feel that it has merit but does not fully meet PLOS ONE’s publication criteria as it currently stands. Therefore, we invite you to submit a revised version of the manuscript that addresses the points raised during the review process.

In addition to addressing the reviewers' content-related suggestions and concerns, please carefully revise language and the formatting issues, which appear to be particularly problematic. Copyediting by a native speaker is strongly recommended.

We would appreciate receiving your revised manuscript by Nov 09 2019 11:59PM. To enhance the reproducibility of your results, we recommend that if applicable you deposit your laboratory protocols in protocols.io, where a protocol can be assigned its own identifier (DOI) such that it can be cited independently in the future. For instructions see: http://journals.plos.org/plosone/s/submission-guidelines#loc-laboratory-protocols

We look forward to receiving your revised manuscript.

Kind regards,

Federica Angeli

Academic Editor

PLOS ONE

Journal Requirements:

2. Under the Methods sections, please provide further information on the source of participants and how they were recruited for this study.

Additional Editor Comments (if provided):

Reviewers' comments:

Reviewer's Responses to Questions

**Comments to the Author**

1. Is the manuscript technically sound, and do the data support the conclusions?

Reviewer #1: Partly

Reviewer #2: Partly

2. Has the statistical analysis been performed appropriately and rigorously? 

Reviewer #1: I Don't Know

Reviewer #2: Yes

3. Have the authors made all data underlying the findings in their manuscript fully available?

Reviewer #1: Yes

Reviewer #2: Yes

4. Is the manuscript presented in an intelligible fashion and written in standard English?

Reviewer #1: No

Reviewer #2: Yes

5. Review Comments to the Author

Reviewer #1: In their paper, Lambooij et al. studied the relative importance, in the eyes of pregnant women, of factors that enhance the quality of maternity care vs the restriction on health provider choice.

Despite addressing a relevant and interesting issue, the paper is overall poorly written, lacks some of the justifications for the choices that were made in the approach, and the presented results might not be fully translatable (due to population selection bias, which is not clearly addressed) and do not seem to fully support the conclusions.

General comments:

-The text lacks much of the punctuation needed to allow for an easy reading, the sentences are often very long, some grammatical mistakes should be corrected and spacing between sections and words should be consistent;

-The English form should be revised, including for typos, verb tenses (eg. in the methods the simple past should normally be used) and verb conjugations inconsistencies;

-Acronyms should all be defined the first time they are used;

-The authors should try to consistently refer to the factors using the same names. This does not seem to happen across the text, in the abstract and in the tables (including, as much as possible, in Table 2);

-the Netherlands is referred to as a country with a presumed high-quality maternity care system. I would not forget that, although in comparison with many countries around the world this is true, perinatal mortality rates in the Netherlands remain among the worst in Western Europe. This is not the topic of this paper, but I would probably very shortly mention it.

Specific comments:

Abstract:

-Look at the general comments.

Introduction:

-Look at general comments;

-End of first paragraph  which heath care system?

-Second sentence of the second paragraph  are the ‘women under the experiment’ the women in the pilots of 2017? Not clear. Please be consistent and clear when referring to the groups of women;

-A central rationale for the study design seems to be the claim that ‘the bundled payment for maternity care negatively affects pregnant women’s freedom of choice’ is this the only difference produced by the implementation of the bundled payment which is coming up in the policy debate (second paragraph of the introduction)? If yes, please clarify and support this claim with references. If not, please explain why you decide to focus mainly on this aspect. This is part of the foundations of this study, the choices made should be well justified;

-In the reviewed evidence (third paragraph introduction) freedom to choose and quality are relevant identified factors. Are those the only one coming out of the literature? If yes, please mention it. If not, what are the others and why do the authors focus only on these 2? the choices made should be well justified;

-Last sentence of first page: the authors claim that ‘this’ (clarify what ‘this’ is) is used to draw conclusions about the extent to which some factors are met under a bundled payment regime. This is not what the study aims at doing, so how can it draw conclusions on it?

Material and methods:

-Look at general comments;

-End of first paragraph: do the authors refer to ‘better-coordinated’ as to ‘quality’? not clear, please be consistent. One of the factors is quality of care, to me, this is not interchangeable with ‘better-coordinated’;

-Since the literature review was conducted before (and to inform) the DCE, I would rather move it before the DCE section;

-in the ‘selection of attributes’ section a pilot study is mentioned. Since this seems to be relevant to justify the list of attributes, it is pertinent to mention something about it. How many women where involved? Were they a good representation of the general population?

-in the recruitment it is mentioned that women who were pregnant for more than 12 weeks were approached. Can the authors clarify why this decision was made?

Results:

-Look at general comments;

-End first paragraph  ‘(23.7% tertiary education in sample vs 14.1% in population)’ I would suggest to change to (23.7% vs 14.1% in tertiary education)’ to be consistent with the other text in the parentheses.

-Equation in bold it looks like another heading.

-Page 10 end of first paragraph  isn’t a third aspect missing?

-Last paragraph ‘about repeated in the first sentence, ‘more’ repeated in the fourth to last sentence.

-Page 10, second to last sentence  ‘this does not imply that respondents…’ why this sentence? The results show an equal distribution of preferences across the 2 factors. I believe to the reader this is already clear, no need to write this.

Discussion:

-Look at general comments;

-Overall only 2 reference are mentioned in the discussion, and all of those in 1 paragraph. I feel a proper discussion with reference to previous literature is missing;

-No need to repeat the aim of the study at the beginning of the discussion, a short summary of the findings is sufficient;

-The first limitation of the study (selection bias, mentioned in the second paragraph on page 12) is not accompanied by a reflection of its implications for the results and possible translation of those into any policy/application;

-Third to last sentence on page 12  ‘Given the limitations…’ in the methods the recruitment criteria mentioned ‘women who were pregnant for >12 weeks’, as such this criteria is the reason for the women recently choosing health care, and this is not due to ‘given the limitations on demographic aspects’ (which per se is not clear what it means). Or is this not clear (so please clarify) and I did not understand?

-Beginning page 13  ‘differ’ …with who?

-First sentence on page 13 what weights? And how are they likely to differ?

-End of first sentence on page 13  ‘the population’ …is this the ‘general population’? Please try to be consistent;

-Page 13, line 9  ‘by testing’ …it’s not clear how this was tested;

-Page 13, last sentence on lines 11 and 12 ‘…and this information is hard to measure…’ not clear what is meant by this sentence. Is it possible to try to rephrase in a clearer way?

-Page 13, line 15 ‘independent organization’ …is this one, more? What type?

-Page 13, third to last sentence ‘Fourth, …’ the third point was already about quality of care. What does this sentence refer to?

-Page 14, first sentence not clear what this means, does this refer to their actual life choices? Or choices in the study? Why not?

-Page 14, second paragraph ‘in this paper…’ to the end of the discussion …why repeating here again a summary of findings? It’s enough at the beginning of the discussion.

Conclusion

-Look at general comments;

-This should not be a summary of the finding again, but a take home conclusion with, possibly, future perspectives;

-Why does the conclusion stresses more the freedom of choice point over the quality? Both factors were studied and both seems equally important, so the manuscript shouldn’t emphasize on over the other, unless there is a clear rationale for that.

Reviewer #2: This paper studies the importance of different maternal care attributes to pregnant women in the Netherlands. The paper is relevant and well written, and definitely deserves a place in an international journal. I have however the following coments, which the authors need to address:

1. The authors refer to previous literature. However, the need for this study is not well elaborated and this should be added at the end of the introduction.

2. In the methods section, details about the sample size calculation, sampling process as well as the process of data collection, should be added.

3. The choice of data analysis is not well motivated and not well specified. Details about the variables, including coding, and the reason for their inclusion should be presented in the method section. Also, equation(s) should be added for illustration, instead of referring to questions in the results section.

4. Results are overall clear except for the equations. It is not clear whether the equations came before the results or vice versa.

5. The discussion section is primarily focused on the discussion of the study design. Authors should extend the discussion and add an interpretation of the results based on previous literature to establish convergent validity and to outline the implications for policy and research.

6. Conclusions refer to women in general, while they need to account for the study context and study sample.

7. There are some typos related to grammar and punctuations. The authors should carefully check the text.

6. PLOS authors have the option to publish the peer review history of their article (what does this mean?). If published, this will include your full peer review and any attached files.

Reviewer #1: Yes: Elena Ambrosino, PhD

Reviewer #2: No

---

## [Author Response · Author response to Decision Letter 0]

9 Nov 2019

Please see the final pages of this resubmission for the document where we marked the comments of the reviewers, and added our responses in text boxes. That will be easier to read than this section, where all marking by text boxes has been removed.

PONE-D-19-23167

Trading freedom of choice for quality of maternity care? A discrete choice experiment amongst pregnant women

PLOS ONE

Dear Dr. Lambooij,

Thank you for submitting your manuscript to PLOS ONE. After careful consideration, we feel that it has merit but does not fully meet PLOS ONE’s publication criteria as it currently stands. Therefore, we invite you to submit a revised version of the manuscript that addresses the points raised during the review process.

Thank you for reviewing the paper and the opportunity to address the point that were raised. In the following we will elaborate on how we addressed the points raised by the reviewers. We have added a substantial text to the introduction to present our case and explain the assumptions behind some of the choices made in the subsequent steps of the study. We have also given more information about data collection. We have restructured the methods section. And we have added substantially to the discussion and conclusion. We (the authors) feel that this has significantly improved the paper. We wish to thank the reviewers and editor for the thorough reading and constructive suggestions that they made. We hope to have 

In addition to addressing the reviewers' content-related suggestions and concerns, please carefully revise language and the formatting issues, which appear to be particularly problematic. Copyediting by a native speaker is strongly recommended.

We acknowledge the need for improvement of the text. We have edited the paper and asked a colleague who lived in the UK for 10 years to critically copyedit the whole manuscript for grammar, typos and unclear sentence structures. The detailed comments of reviewer 1 also helped us to solve a number of language problems. If desired by the editorial board we will additionally send the manuscript to an native speaking editor.

We would appreciate receiving your revised manuscript by Nov 09 2019 11:59PM. We do not wish the make changes to the financial disclosure.

To enhance the reproducibility of your results, we recommend that if applicable you deposit your laboratory protocols in protocols.io, where a protocol can be assigned its own identifier (DOI) such that it can be cited independently in the future. For instructions see: http://journals.plos.org/plosone/s/submission-guidelines#loc-laboratory-protocols

• A rebuttal letter that responds to each point raised by the academic editor and reviewer(s). This letter should be uploaded as separate file and labeled 'Response to Reviewers'.

• A marked-up copy of your manuscript that highlights changes made to the original version. This file should be uploaded as separate file and labeled 'Revised Manuscript with Track Changes'.

• An unmarked version of your revised paper without tracked changes. This file should be uploaded as separate file and labeled 'Manuscript'.

We look forward to receiving your revised manuscript.

Kind regards,

Federica Angeli

Academic Editor

PLOS ONE

Journal Requirements:

We have changed headings and files names after closely examining the guidelines

2. Under the Methods sections, please provide further information on the source of participants and how they were recruited for this study.

We gave more information on the recruitment process of the respondents in the section “Recruitment”. The section now reads: 

We used the database of a panel that covers 75% of all women who ever gave birth and/or who are pregnant in the Netherlands. Pregnant women in the Netherlands can sign up to receive “The happy box”, a box that contains number of items such as cream, a pacifier, and clothing, that are useful to care for a new-born baby. The happy box is organised by a large number of collaborative stores. Many women apply for this box and are subsequently included in this database. For this study, all registered women who were pregnant for more than 12 weeks at the time of data collection were approached. 

Could you please advice us on how to best present our data? In the current revision, we added S4 and S5, being a csv-file of the data we used and a codebook with labels. If you decide to accept the paper, the data can be added to the publication as an Supplementary file. However, if you prefer to publish the data in another way, could you please instruct us on how this can be done?

We renamed the additional information to Supporting files 1-3 and changed the references in the manuscript accordingly.

Reviewers' comments:

Reviewer's Responses to Questions

Comments to the Author

1. Is the manuscript technically sound, and do the data support the conclusions?

Reviewer #1: Partly

Reviewer #2: Partly

2. Has the statistical analysis been performed appropriately and rigorously? 

Reviewer #1: I Don't Know

Reviewer #2: Yes

3. Have the authors made all data underlying the findings in their manuscript fully available?

Reviewer #1: Yes

Reviewer #2: Yes

4. Is the manuscript presented in an intelligible fashion and written in standard English?

Reviewer #1: No

Reviewer #2: Yes

5. Review Comments to the Author

Reviewer #1: In their paper, Lambooij et al. studied the relative importance, in the eyes of pregnant women, of factors that enhance the quality of maternity care vs the restriction on health provider choice.

Despite addressing a relevant and interesting issue, the paper is overall poorly written, lacks some of the justifications for the choices that were made in the approach, and the presented results might not be fully translatable (due to population selection bias, which is not clearly addressed) and do not seem to fully support the conclusions.

We thank the reviewer for critically reading the paper and providing us with constructive and detailed comments. In de the following we address the points that were raised and explain how we tried to deal with them.

We have edited the paper and asked a colleague who lived in the UK for 10 years to critically review the whole manuscript for grammar, typos and unclear sentence structures. If desired by the editorial board we will additionally send the manuscript to an editor.

General comments:

-The text lacks much of the punctuation needed to allow for an easy reading, the sentences are often very long, some grammatical mistakes should be corrected and spacing between sections and words should be consistent;

Please see our answer to the prior comment.

-The English form should be revised, including for typos, verb tenses (eg. in the methods the simple past should normally be used) and verb conjugations inconsistencies;

Please see our answer to the first comment.

-Acronyms should all be defined the first time they are used;

This was indeed done inconsistently. The paper was scanned for acronyms and definitions and explanations were added throughout the paper.

-The authors should try to consistently refer to the factors using the same names. This does not seem to happen across the text, in the abstract and in the tables (including, as much as possible, in Table 2);

Thank you for your concise checking of the paper. In response to this observation we checked the abstract and result sections on the names of the factors in the DCE. We renamed the factors in the text according to the naming in table 2.

In track changes all changes are marked.

-the Netherlands is referred to as a country with a presumed high-quality maternity care system. I would not forget that, although in comparison with many countries around the world this is true, perinatal mortality rates in the Netherlands remain among the worst in Western Europe. This is not the topic of this paper, but I would probably very shortly mention it.

The reviewer is correct in this remark. In fact, the relatively high infant mortality was an important driver for the restructuring of the health care delivery in the Netherlands. 

Infant mortality in the Netherlands was relatively high in 2000 and decreased broadly in line with other countries until 2008. Subsequently, this progress has continued and the Netherlands has outperformed many similarly developed countries. Since 2000, infant mortality has decreased by nearly 39% and stillbirths have decreased by 66%. A national strategy is assumed to contribute significantly to the increase in performance of maternity care. 

(please see Struijs, J. N. and D. S. Hargreaves. "Turning a Crisis into a Policy Opportunity: Lessons Learned So Far and Next Steps in the Dutch Early Years Strategy." Lancet Child Adolesc Health 3, no. 2 (Feb 2019): 66-68. https://dx.doi.org/10.1016/s2352-4642(18)30384-5. For more details)

Initially we added the following line in the final section of the paper:

“The introduction of Bundled payment in maternity care is a recent development of a number of improvements of the Dutch maternity care, coinciding with perinatal mortality rates becoming more similar to other high income countries [40].”

However, during the final editing round, the section appeared to be artificially added to the paper. Given the reference in this reply, we acknowledge that the Netherlands still has a relatively high level for perinatal mortality, but that improvements have been made. We still believe that compared to the rest of the world the Dutch health care system, including maternity care is of high quality. We therefore propose to acknowledge the observation of the reviewer, but to not add this or a similar text to the paper.

Specific comments:

Abstract:

-Look at the general comments.

Please see our first response on editorial activities.

Introduction:

-Look at general comments;

-End of first paragraph  which heath care system?

This was indeed unclear. In response to other points raised by the reviewer, we added a section prior to this section elaborating on Bundled payment an a number of developments specific to the Dutch system. We expect that it becomes clearer to the readers that we refer to the Dutch system. To avoid all risk of misunderstanding, we added the word “Dutch”:

[…. the Dutch Health care system]

-Second sentence of the second paragraph  are the ‘women under the experiment’ the women in the pilots of 2017? Not clear. Please be consistent and clear when referring to the groups of women;

The reviewer is correct that this in unclear. This refers to women living in the Netherlands at the time of the experiment. We changed the text to:

“In Dutch bundled payment pilots that started in 2017, pregnant women are still able to choose their providers themselves by the introduction of the “bundle breaker”. In practice this meant that women could receive service delivery from care providers whom were not subcontracted by the IMCO if they wished to. .”

-A central rationale for the study design seems to be the claim that ‘the bundled payment for maternity care negatively affects pregnant women’s freedom of choice’ is this the only difference produced by the implementation of the bundled payment which is coming up in the policy debate (second paragraph of the introduction)? If yes, please clarify and support this claim with references. If not, please explain why you decide to focus mainly on this aspect. This is part of the foundations of this study, the choices made should be well justified;

Initially we hoped to provide sufficient clarification for our choices by means of the information in S1. However, the reviewer is correct that this is a central assumption in our paper and should be included in the introduction. 

We prefer to still include supplementary file 1 (S1) for a technical description of the organisation of care in a bundled payment regime followed by a description of the organization of maternity care in the Bundled payment system, including a number of references. 

In reaction to this comment, we added the following text to the introduction, to justify the claim that bundled payment may improve quality of care and simultaneously may result in reduced freedom to choose. The added text is as follows:

“The rationale behind payment models such as bundled payment is that they incentivize care coordination, and consequently stimulate the use of high-quality care by increased provider accountability. Empirical associations have been found between use of integrated care and increased patient satisfaction, perceived quality of care patient access to services [8]. Use of Bundled Payment is a shift from the commonly used fee-for-service models in which the financial risks are primarily borne by the payers. A fee-for-service system supposedly enables providers to deliver larger volumes of care regardless of the quality of their services [9-13]. Bundled payment models shift the incentives towards health outcomes and health spending, motivating providers to deliver high quality of care [10]. Bundled payment serves as an incentive to avoid unnecessary care, and encourages more efficient coordination between care practitioners. In the Netherlands Bundled Payment for diabetes care had been introduced in 2007 and had both positive and negative consequences. Positive consequences included quality improvements in care delivery processes and in the transparency of delivered care. Negative consequences included antitrust concerns and more limited choice for patients [14].”

-In the reviewed evidence (third paragraph introduction) freedom to choose and quality are relevant identified factors. Are those the only one coming out of the literature? If yes, please mention it. If not, what are the others and why do the authors focus only on these 2? the choices made should be well justified;

The most important reason for this study is the policy debate on limitations to freedom to choose as a result of integrated care organizations in maternity care. A number of these factors are related to the mother and cannot be affected by the care delivery organization (e.g. previous experiences). 

Given this comment we realize that the introduction may be written too compactly. No, these are not the only two factors that are found to be important, and yes these findings stem from our literature search. In Supplementary file S2 we list all factors that are mentioned in the literature that we found in preparation of the DCE. 

We added the following paragraph to illustrate finding from the literature we find most relevant, and refer to the list in S2 to point out that the complete list of factors that are relevant in choices for maternity care is longer. 

The paragraph now reads:

“In the considerable body of evidence on preferences of pregnant women on maternity care, aspects related to provider choice [8-14] and aspects of maternity care related to quality of care [11-13, 15-18] are identified to be relevant. But also personal experiences [12, 14, 15, 19-21], birth place location [13, 21-23], and birth place process [10] have been found to affect the preference of pregnant women. Pain relief is found to be important by both the women and their partners and in turn this is influenced by previous experiences [24]. Also previous DCEs have found that women value nurses appointments during being pregnant [18] and that women are willing to accept fewer visits by maternity care professionals, if this is replaced with sufficient alternative care [17]. Supplementary file S2 provides a table with all factors affecting pregnant women’s preferences which the study identified. In this study we focussed on aspects that are indications of or affect quality of care and provider choice. “

-Last sentence of first page: the authors claim that ‘this’ (clarify what ‘this’ is) is used to draw conclusions about the extent to which some factors are met under a bundled payment regime. This is not what the study aims at doing, so how can it draw conclusions on it?

Thank you for the opportunity to clarify this point. “This” refers to the results of the DCE, stating the preference structures we found. We made this explicit in the text: 

“We used the preference structures that resulted from the DCE to draw conclusions about the extent to which women’s personal preferences concerning quality of maternity care and choice in provider are sufficiently met under a bundled payment regime.”

Material and methods:

-Look at general comments;

Please see our first response on editorial activities.

-End of first paragraph: do the authors refer to ‘better-coordinated’ as to ‘quality’? not clear, please be consistent. One of the factors is quality of care, to me, this is not interchangeable with ‘better-coordinated’;

The reviewer is correct in the fact that better coordination and better quality of care are not interchangeable. 

In maternity care, midwives and gynecologists have to work together closely to ensure good quality health care. Gynecologists are based in hospitals while many midwives work in other organizations. This makes the need for interorganizational cooperation clear immediately. The relationship between use of integrated care and quality of care is found in the literature (Baxter et al 2018). 

In response to both reviewer 1 and reviewer 2, we added text to the introduction of the paper to explicate the mechanism behind the coordination and quality of care We hope to clarify the relationship to the reader and justify our choices. In the remainder of the text we will focus on quality of care, since this is the most important driver behind the implementation of integrated care models.

1. Baxter S, Johnson M, Chambers D, Sutton A, Goyder E, Booth A. The effects of integrated care: a systematic review of UK and international evidence. BMC health services research. 2018;18(1):350.

-Since the literature review was conducted before (and to inform) the DCE, I would rather move it before the DCE section;

If one considers the chronological process, the reviewer is absolutely correct. In fact, in earlier versions of the paper, the literature study was presented before the DCE to represent the chronological order of the steps. However, in (internal) review processes, this was found to be confusing to many readers and repeatedly the reviewers argued that the rationale for the literature study was to identify the relevant attributes for the DCE. And when we presented the literature study before the DCE, the focus shifted. 

We then decided that it was most important to justify how and why the DCE was designed as it eventually turned out and have the goal of the literature study be guiding for the place it would get in the paper. Since it is a first step in deciding which attributes to include, we decided to present as such.

-in the ‘selection of attributes’ section a pilot study is mentioned. Since this seems to be relevant to justify the list of attributes, it is pertinent to mention something about it. How many women where involved? Were they a good representation of the general population?

The reviewer is correct that we need to report relevant aspects of the pilot study.

In the text we mention that ten women were included. They were approached randomly on site and given the context we did not ask them their age or educational level.

“In total, ten pregnant women completed a pilot questionnaire while also discussing their opinions. In a midwifery practice we randomly asked ten pregnant women to fill out the questionnaire and give feedback. The midwifery practice was located in the center of a medium-sized city”

-in the recruitment it is mentioned that women who were pregnant for more than 12 weeks were approached. Can the authors clarify why this decision was made?

Thank you for the opportunity to clarify. In the Netherlands it is custom that women will not tell others that they are pregnant after week 12 of the pregnancy. Because of this, we assumed that it would be merely impossible to find women who were less than 12 weeks pregnant.

Results:

-Look at general comments;

Please see our first response on editorial activities.

-End first paragraph  ‘(23.7% tertiary education in sample vs 14.1% in population)’ I would suggest to change to (23.7% vs 14.1% in tertiary education)’ to be consistent with the other text in the parentheses.

We agree with the comment of the reviewer and changed the text accordingly.

-Equation in bold it looks like another heading.

Thank you. The editor asked us to use heading in accordance with the journal guide lines. That resolved this comment as well.

-Page 10 end of first paragraph  isn’t a third aspect missing?

The reviewer is correct, thank you. We added the third aspect to the sentence:

“Next, the three aspects related to quality were weighed: working form a single EMR (RIS=0.51), advice of friends (RIS=0.17), and the information given by the midwife (RIS=0.10).”

-Last paragraph ‘about repeated in the first sentence, ‘more’ repeated in the fourth to last sentence.

We changed this for the double ‘about’:

“Summarizing, we found four reference structures, each with about equal CAPs (Table 2, bottom row).”

The double more is intended: the first more refers to ‘choice’, and the second to ‘highly’. Removing one alters the meaning of the sentence

-Page 10, second to last sentence  ‘this does not imply that respondents…’ why this sentence? The results show an equal distribution of preferences across the 2 factors. I believe to the reader this is already clear, no need to write this.

We deleted this sentence.

Discussion:

-Look at general comments;

We edited the text in response to this and previous requests.

-Overall only 2 reference are mentioned in the discussion, and all of those in 1 paragraph. I feel a proper discussion with reference to previous literature is missing;

Given other responses of this reviewer and reviewer 2, substantial sections of the discussion are rewritten. In the process, more references were added. We hope to have addressed this comment satisfactorily by the added and changed text fragments.

-No need to repeat the aim of the study at the beginning of the discussion, a short summary of the findings is sufficient;

We deleted the aim in the beginning of the discussion. 

-The first limitation of the study (selection bias, mentioned in the second paragraph on page 12) is not accompanied by a reflection of its implications for the results and possible translation of those into any policy/application;

The reviewer is correct that we did not relate this to policy implications. We believe that our results are valuable to the policymakers, given this restriction.

“We have therefore no reason to assume that the relative importance of the attributes in our results differ from the relative importance in real life choices, when just looking at ranking of the attributes. However the precise parameter values are likely to differ between the sample in this study and the general population. ”

We therefore believe that the practical implications that we mention are not affected by this limitation.

-Third to last sentence on page 12  ‘Given the limitations…’ in the methods the recruitment criteria mentioned ‘women who were pregnant for >12 weeks’, as such this criteria is the reason for the women recently choosing health care, and this is not due to ‘given the limitations on demographic aspects’ (which per se is not clear what it means). Or is this not clear (so please clarify) and I did not understand?

Thank you again for your thorough reading and comments. This part of the paragraph refers to the demographic aspect “educational level”, the topic of the text above. Yes, the quality of the sample is discussed based on the aspect of educational level and the fact that it includes women who are pregnant when they fill in the questionnaire.

We rewrote the paragraph to present our case more clearly

“Despite the overrepresentation of highly educated respondents, the sample benefits from the fact that all respondents had actual experience in the choice in the DCE because they were pregnant at the time of filling in the questionnaire. This meant that respondents were familiar with and had experience in the decisions they were asked to make in DCEs.”

-Beginning page 13  ‘differ’ …with who?

This refers to a possible difference between the relative importance of the attributes in our results to the real life relative importance of the choices of pregnant women for maternity care providers.

We changed the text:

“We have therefore no reason to assume that the relative importance of the attributes in our results differ from the relative importance in real life choices, when just looking at the ranking of attributes. However the precise parameter values are likely to differ between the sample in this study and the general population [40].”

-First sentence on page 13 what weights? And how are they likely to differ?

We mean to refer to the size of the parameters. The reviewer is correct that this may be an unfit term to use in this context. |We therefore rephrased it to “parameter values”: 

“However the precise parameter values are likely to differ between the sample in this study and the general population.”

-End of first sentence on page 13  ‘the population’ …is this the ‘general population’? Please try to be consistent;

The reviewer is correct. We added “general”.

-Page 13, line 9  ‘by testing’ …it’s not clear how this was tested;

We changed this to “by conducting a pilot study on the questionnaire,”.

-Page 13, last sentence on lines 11 and 12 ‘…and this information is hard to measure…’ not clear what is meant by this sentence. Is it possible to try to rephrase in a clearer way?

This refers to the information on the relative importance as stated in the first part of the sentence. We elaborated more on this point, aiming to express it more clearly

“Nonetheless, by conducting a pilot study on the questionnaire and involving experts in the development of the scenario’s, it is assumed that the five most important attributes are included in the DCE. A distinctive feature of a DCE is that it provides information on the relative importance of the included factors [26].This information is hard to measure with other research instruments, while in real life, similar trade-offs are an intrinsic fact of almost every choice. We therefore believe that for policymakers, the current DCE can be used to understand the most important factors that women find important when choosing maternity care services.”

-Page 13, line 15 ‘independent organization’ …is this one, more? What type?

This may be too specific information to deal with in the paper. It refers to the fact that information on quality of care might be manipulated. If information of quality of health care services is provided, this needs to be given by an organization that holds no interest in the success of the maternity care providers. As such, if information of the quality of the maternity care providers is given, this organization needs to be independent from the maternity care organization. It is unimportant what type of organization this is, assuming that it has the capability to provide relevant and accurate information on the performance of the maternity care organization.

-Page 13, third to last sentence ‘Fourth, …’ the third point was already about quality of care. What does this sentence refer to?

This indeed was unclear. Thank you for pointing it out. The third point is about the operationalization of quality, the fourth point is about the question whether pregnant women in real life would have access to information about the quality of maternity care services.

We added the following sentence to explicate the difference.

“This means that in the current study, the respondents are likely to have more information on factors that affect the quality of maternity care than in real life.”

-Page 14, first sentence not clear what this means, does this refer to their actual life choices? Or choices in the study? Why not?

Yes, this refers to the real life practice. We made this more explicit:

“This implies that even though this study yields insights into the preferences of women using maternal care, in practice they may not be able to realise their preferences due to lack of information regarding the real-life organisation of healthcare providers. “

-Page 14, second paragraph ‘in this paper…’ to the end of the discussion …why repeating here again a summary of findings? It’s enough at the beginning of the discussion.

The reviewer is correct. We deleted this part if the text and restructured it to present our point about the implication clearly:

“Considering the practice of maternity care organized via bundled payment arrangements, it appears that the limitations in provider choice may be acceptable for the (future) clients of maternity care. All identified groups of women pay attention to the possibilities of provider choice, and on average more choice is preferred over less choice, but complete provider choice appears to be less attractive than choosing for a bundle of care if it contains to opportunity to change one of the maternity care providers. This may be due to the fact that of complete provider choice implies that women also need to find information about all health care providers involved, which is more costly than acquiring information for making one or two choices.”

Conclusion

-Look at general comments;

Please see our first response on editorial activities.

-This should not be a summary of the finding again, but a take home conclusion with, possibly, future perspectives;

Both reviewers 1 and 2 make suggestions to improve the discussion of the paper. We tried to combine all suggestions by rewriting the section on conclusions:

“Both quality of care and provider choice aspects are important to pregnant women when choosing for maternity care. Reorganizing health care practices using Bundled payment to encourage cooperation between health care practitioners was already known to entail advantages and disadvantages. And limited provider choice is known to be a disadvantage [13]. Results from this study indicate that a substantial part of the clients or patients of health care services are willing to accept this disadvantage and that another part of the clients attach great value to more provider choice. 

For policy makers and practitioners this study revealed that not limiting provider choice under a bundled payment model is essential to deliver maternity care that meets the demands of all pregnant women. In order to realize support among pregnant women for a bundled payment model, these payment models should include unlimited provider choice.”

-Why does the conclusion stresses more the freedom of choice point over the quality? Both factors were studied and both seems equally important, so the manuscript shouldn’t emphasize on over the other, unless there is a clear rationale for that.

The section starts with the conclusion “Both quality of care and freedom of choice aspects are important to pregnant women….”. Also in the Discussion section we state that about half of the women find provider choice most important. In the revised introduction, we now explicitly state that the positive side of BP is better of quality of care and the downside is the reduced freedom to choose providers. 

The point we try to make in the discussion is that policy makers need to take account of the different preferences in the population of pregnant women when they strive to organize maternity care in such a way that it meets the demands of all women. This means that they need to organize the health care system in such a way that an important disadvantage of BP is dealt with.

Given the changes in the introduction and the altered text in the final section, we hope to have taken away the suggestion that we stress quality over freedom of choice. 

The final sentences now read:

“For policymakers and practitioners this study revealed that not limiting provider choice under a bundled payment model is essential to delivering maternity care that meets the demands of all pregnant women. In order to realize support for pregnant women in a bundled payment model, these payment models should include unlimited provider choice.” 

Reviewer #2: This paper studies the importance of different maternal care attributes to pregnant women in the Netherlands. The paper is relevant and well written, and definitely deserves a place in an international journal. I have however the following comments, which the authors need to address:

We thank the author for the constructive comments. In the following sections we address them point by point.

1. The authors refer to previous literature. However, the need for this study is not well elaborated and this should be added at the end of the introduction.

We rewrote the introduction, referring to changes in health care policies and the influence it may have on maternity care. We also added more introduction to evidence on the preferences of pregnant women. We hope that this sections assist in explaining the need for this paper. Specifically in response to this point , we added the following section to the introduction:

“The results of this study are relevant for policy makers, as it will give them information on the preferences of the receivers of maternity care, pregnant women. The results will show whether it is necessary to maintain the maximum level of provider choice for pregnant women, or whether women are willing to accept (some) limitations to freedom to choose. The study adds information to the existing body of literature on the willingness of pregnant women to trade between different aspects of maternity care, providing more information on the relative importance of different aspects of maternity care, as experienced by pregnant women.”

By this text, in combination with the more extensive introduction on Bundled Payment in the, in response to this reviewer and reviewer 1, we hope that we addressed the reviewers’ comment satisfactorily.

2. In the methods section, details about the sample size calculation, sampling process as well as the process of data collection, should be added.

Indeed we can be more explicit about our considerations in gathering the data. We added the following text on sampling process and data collection:

“We used the database of a panel that covers 75% of all women who ever gave birth and/or who are pregnant in the Netherlands. Pregnant women in the Netherlands can sign up to receive “The happy box”, a box that contains number of items such as cream a pacifier and clothing, that are useful to care for a new-born baby. The happy box is organised by a large number of collaborative stores. Many women apply for this box and are subsequently included in this database. For this study, all registered women who were pregnant for more than 12 weeks at the time of data collection were approached by email. The email contained an introductory text to explain the aim of the study and a link to the questionnaire. All women were emailed once, no reminders were sent. Women were able to stop filling in the questionnaire at any time. Only the questionnaires that were filled in completely were included in the analyses.”

A priori sample-size calculations represent a challenge in DCE experiments. Most published choice experiments have a sample size of 100 to 300 respondents (Marshall et al., 2010). However, minimum sample size depends on several criteria, including the question format, the complexity of the choice task, the desired precision of the results, and the need to conduct subgroup analyses (De Bekker-Grob et al, 2015). De Bekker-Grob (2015) reports in a paper on sample size: that there is no analytic solution or power calculation that can be used to determine the appropriate sample size for a DCE unless the researcher has enough information to inform the selection of priors.

By using a broader generally accepted rule of thumb, sample sizes are deemed sufficient as described below.

 Samplesize>500l/TA

Sample size depends on the number of choice tasks (T), the number of alternatives in a choice set (A), and largest number of levels in any attribute (l). In our case a DCE needs at least 84 respondents to estimate the main effects. Given proposed latent class analysis (including heterogeneity) and blocked design (asking people to only complete a subset of the choice tasks, thereby limiting the burden on every single participant) we would need to have 504 (84*3blocks*2(model adjustment)) respondents. According to these calculations a sample size of 600 respondents should be sufficient to answer the proposed research questions and provide enough information to identify preferences in these groups and comparisons across these groups with acceptable precision.

Marshall, D., J. F. Bridges, B. Hauber, R. Cameron, L. Donnalley, K. Fyie, and F. R. Johnson. "Conjoint Analysis Applications in Health - How Are Studies Being Designed and Reported?: An Update on Current Practice in the Published Literature between 2005 and 2008." Patient 3, no. 4 (Dec 1 2010): 249-56. https://dx.doi.org/10.2165/11539650-000000000-00000.

de Bekker-Grob, E. W., B. Donkers, M. F. Jonker, and E. A. Stolk. "Sample Size Requirements for Discrete-Choice Experiments in Healthcare: A Practical Guide." Patient 8, no. 5 (Oct 2015): 373-84. https://dx.doi.org/10.1007/s40271-015-0118-z.

We added the following text to the paper to justify our choices.

“To estimate the required sample size, we used a generally accepted rule of thumb

 Sample size > 500l / TA

Where T= the number of choice tasks, A= the number of alternatives in a choice set, and l= largest number of levels in any attribute (l). In our case a DCE needs at least 84 respondents to estimate the main effects. Given proposed latent class analysis (including heterogeneity) and blocked design (asking people to only complete a subset of the choice tasks, thereby limiting the burden on every single participant) we estimated to need 504 (84*3blocks*2(model adjustment)) respondents. ”

3. The choice of data analysis is not well motivated and not well specified. Details about the variables, including coding, and the reason for their inclusion should be presented in the method section. Also, equation(s) should be added for illustration, instead of referring to questions in the results section.

The reviewer is correct. 

We restructured the analyses section, moving the equations to the analysis section. References to the variables in the equations to the tables were removed. 

4. Results are overall clear except for the equations. It is not clear whether the equations came before the results or vice versa.

The reviewer is correct that this is unclear. 

The equations were constructed prior to the data collection since they are defined by attribute and level selection. The class assignment model was determined based on model fit, after collecting the data. 

The reviewer indicates that the current position is confusing to readers. We therefore moved the equations to the method section. We believe that the complete model is best understood when the utility equations and class assignment model are presented it their proximity. For now we also moved the class assignment model to the methods. However, if the editor or reviewer wishes to present the class assignment model in the results, we will do so.

5. The discussion section is primarily focused on the discussion of the study design. Authors should extend the discussion and add an interpretation of the results based on previous literature to establish convergent validity and to outline the implications for policy and research.

We thank the reviewer for the opportunity to present our ideas on the implications of our work. We added a number of texts to address this point by the reviewer.

When discussing the methodological issues, we added the following fragments to indicate what we see to be valuable future research.

“Future studies may yield relevant new information on the relative importance of harder to affect aspects such as previous experiences of the women compared to factors that can be affected by maternity care providers or policymakers.”

“Measuring quality of care is difficult and depending on the study’s aim, different angles can be chosen [e.g. 42, 43, 44]. In our case, we needed to exclude more medical technical measures of quality because our main focus was to resemble a choice similar to a real-life choice as much as possible. When it comes to quality of care, it may be useful to compare the relative importance of medical technical measures of quality of care and more patient- centred measures of quality of care in future studies.”

“These results are in line with previous findings that women are willing to trade between factors of maternity care [24]. What this study adds to the literature is the diversification in the population of interest: different women will make different choices when they can. Some pregnant women are likely to opt for fewer choice in provider, based on the assumption that they would receive better quality of care, while others will pass on the presumed quality of care improvement to maintain their ability to choose their providers freely. 

This study particularly focussed on factors of maternity care that can be influenced by healthcare providers and policymakers. In this choice experiment, we excluded personal experiences [22, 38] women’s beliefs and values [23] or type of birth setting [16, 17, 21], even though these factors are known to be relevant to affect the preferences of pregnant women. The downside of restricting this study to factors that can be affected by the organisation of care is that we do not know the relative importance of those factors compared to the factors we did not include. However, the advantage is that this study provides practitioners and policymakers with relevant information on how maternity care clients will respond to changes in how their maternity care is organised. Future studies may yield relevant new information on the relative importance of harder-to-affect aspects such as women’s previous experiences compared to factors that can be affected by maternity care providers or policymakers.”

We also added notions on policy implications:

“Considering the practice of maternity care provided through bundled payment arrangements, it appears that the limitations in provider choice may be acceptable for (future) clients of maternity care. All identified groups of women consider their ability to choose a provider, and on average more choice is preferred over less choice, but being able to choose any provider appears to be less attractive than choosing a bundle of care containing opportunities to change one of the maternity care providers. This may be due to the fact that being able to choose any provider implies that women also need to find information about all possible healthcare providers, which requires more effort than acquiring information for making one or two choices.”

“Reorganizing healthcare practices using Bundled payment to encourage cooperation between healthcare practitioners was already known to entail advantages and disadvantages. Limited provider choice is known to be a disadvantage [14]. Results from this study indicate that a substantial number of clients or patients are willing to accept this disadvantage and that other clients attach greater value to more choice in providers. 

For policymakers and practitioners this study revealed that not limiting provider choice under a bundled payment model is essential to delivering maternity care that meets the demands of all pregnant women. In order to realize support for pregnant women in a bundled payment model, these payment models should include unlimited provider choice. ”

6. Conclusions refer to women in general, while they need to account for the study context and study sample.

Both reviewers 1 and 2 make suggestions to improve the discussion of the paper. We tried to combine all suggestions by rewriting the section on conclusions:

“Both quality of care and provider choice aspects are important to pregnant women when choosing for maternity care. Reorganizing health care practices using Bundled payment to encourage cooperation between health care practitioners was already known to entail advantages and disadvantages. And limited provider choice is known to be a disadvantage [13]. Results from this study indicate that a substantial part of the clients or patients of health care services are willing to accept this disadvantage and that another part of the clients attach great value to more provider choice. 

For policy makers and practitioners this study revealed that not limiting provider choice under a bundled payment model is essential to deliver maternity care that meets the demands of all pregnant women. In order to realize support among pregnant women for a bundled payment model, these payment models should include unlimited provider choice.”

7. There are some typos related to grammar and punctuations. The authors should carefully check the text.

We re-read the text ourselves and asked a colleague who lived in the UK for ten years and was not involved in the project to also correct the text for grammar and punctuation. Additionally, in our responses to reviewer 1, we addressed a number of particular mistakes pointed out to us by that reviewer. By these actions we hope to have resolved this point.

6. PLOS authors have the option to publish the peer review history of their article (what does this mean?). If published, this will include your full peer review and any attached files.

Do you want your identity to be public for this peer review? For information about this choice, including consent withdrawal, please see our Privacy Policy.

Reviewer #1: Yes: Elena Ambrosino, PhD

Reviewer #2: No

---

## [Decision Letter · Decision Letter 1]

16 Dec 2019

PONE-D-19-23167R1

Trading patients’ choice in providers for quality of maternity care? A discrete choice experiment amongst pregnant women

PLOS ONE

Dear Dr. Lambooij,

Thank you for submitting your manuscript to PLOS ONE. After careful consideration, we feel that it has merit but does not fully meet PLOS ONE’s publication criteria as it currently stands. Therefore, we invite you to submit a revised version of the manuscript that addresses the points raised during the review process.

We would appreciate receiving your revised manuscript by Jan 30 2020 11:59PM. To enhance the reproducibility of your results, we recommend that if applicable you deposit your laboratory protocols in protocols.io, where a protocol can be assigned its own identifier (DOI) such that it can be cited independently in the future. For instructions see: http://journals.plos.org/plosone/s/submission-guidelines#loc-laboratory-protocols

We look forward to receiving your revised manuscript.

Kind regards,

Federica Angeli

Academic Editor

PLOS ONE

Reviewers' comments:

Reviewer's Responses to Questions

**Comments to the Author**

1. If the authors have adequately addressed your comments raised in a previous round of review and you feel that this manuscript is now acceptable for publication, you may indicate that here to bypass the “Comments to the Author” section, enter your conflict of interest statement in the “Confidential to Editor” section, and submit your "Accept" recommendation.

Reviewer #1: All comments have been addressed

Reviewer #2: (No Response)

2. Is the manuscript technically sound, and do the data support the conclusions?

Reviewer #1: Yes

Reviewer #2: Partly

3. Has the statistical analysis been performed appropriately and rigorously? 

Reviewer #1: Yes

Reviewer #2: Yes

4. Have the authors made all data underlying the findings in their manuscript fully available?

Reviewer #1: Yes

Reviewer #2: Yes

5. Is the manuscript presented in an intelligible fashion and written in standard English?

Reviewer #1: Yes

Reviewer #2: Yes

6. Review Comments to the Author

Reviewer #1: Thank you to Labooij and co-authors for the revised version of their manuscript. Following their effort, the quality and clarity of the article have improved considerably.

I have few additional minor remarks that I feel should still be addressed before publication.

General comment:

Thought the form is significantly better (with the exception of the discussion which seems of lesser quality), I would suggest an additional review to check for punctuation, spacing between words and words/punctuation, and for form (eg: Bundled Payment, with capitals or not? be consistent).

Specific comments (refer to the clean version of the manuscript, not the track/trace) (words in capital letters are my suggestions for addition):

Abstract:

-if possible (if words allow) I would contextualize the background to the Netherlands. Eg: ‘…describes a DUTCH study…’

-third line ‘….pregnant women’s….’ NOT ‘…[THE] pregnant women’s…’

Introduction:

-the second sentence of the second paragraph (starting with ‘Empirical…) is not grammatically correct

-last sentence in the first page, I would specify to: ‘in THE six DUTCH regions…’

-second page of the introduction, second paragraph, sentence starting with ‘But also…’: is ‘birth place process’ the right term? Also, is reference #10 the correct one? I couldn’t find any info on birth place process in reference #10

-last paragraph second page, ‘the results of this study are relevant for policymakers, as THEY PROVIDE’ and not it provides

Results:

-Sample section, last sentence: ‘…and women who were pregnant with their…  should this be: ‘…and MORE women who were pregnant with their first child….'?

Discussion:

-second sentence, I would suggest: ‘For SOME women the unlimited provider choice is the most important aspect, and THEY find this more….’

-first sentence second paragraph, remove the A before about: ‘…that people had [a] about a one in four….’

-first sentence second page: ‘….difficult-to-measure (add the dash between to and measure) aspects such as [of] harder-to-measure…’ AND try not to repeat ‘such as’ in the same sentence so closely

-last paragraph second page, first sentence: remove the last word ‘attributes’, you already mentioned it earlier in the sentence, that is enough

-last line second page: scenario’s is no plural in English (that is the Dutch form), in English it’s scenarios

-third page, third line: ‘intrinsic factOR’

-end of second paragraph: ‘Since DCEs’ main objectives ARE to describe…’ OR ‘Since DCEs’ main OBJECTIVE is to describe…’

-third paragraph: third and fourth sentence start with ‘this means’…it feels you are trying to explain a concepts with several sentences without succeeding. Can you limit the use of ‘this means’ to once?

-third page last sentence: ‘….in practice THE WOMEN...’  instead of ‘they’ I would write ‘the women’, since the subject in the first part of the sentence is the study, not the women.

-fourth page, twice used: ‘medical technical measures’  is this the right term? It was not used in the text before and it’s not an obvious term

-fourth page, third line: I would remove ‘choice similar to a’

-the very last sentence of the discussion seems to be speculation without references. It’s a peculiar way to end a discussion. I would revise it

Conclusion:

-usually reference are not used in the conclusion, is references 14 needed there again?

Reviewer #2: Authors have made an effort to address my comments. However, the following comments need additional attention.

• Details about the variables, including coding, and the reason for their inclusion should be presented in the methods section. Supporting file S5 Codebook to dataset is fine but it is important to provide the above information in the methods section.

• Authors should extend the discussion (the first 3 paragraphs) and add an interpretation of the results based on previous literature to establish convergent validity and to outline the implications for policy and research. I mean an interpretation of the findings on women’s preferences not the methodology aspects of the study. Why are these preferences observed? How do they compare to previous studies? What do they mean for policy and research?

• The selection bias and the limitations of it should be discussed more substantially to establish the possibility to extrapolate the results. Unless this is done, conclusions only refer to women in the register “The happy box” and not to all women in the Netherlands. The representativeness of the sample should be accounted for when making conclusions. In this regard, the validity of the current conclusions is unclear.

7. PLOS authors have the option to publish the peer review history of their article (what does this mean?). If published, this will include your full peer review and any attached files.

Reviewer #1: Yes: Elena Ambrosino

Reviewer #2: No

---

## [Author Response · Author response to Decision Letter 1]

9 Mar 2020

Please see the final pages of this resubmission for the same comments and responses, but with lay out for easier reading.

Reviewers' comments:

Reviewer's Responses to Questions

Comments to the Author

1. If the authors have adequately addressed your comments raised in a previous round of review and you feel that this manuscript is now acceptable for publication, you may indicate that here to bypass the “Comments to the Author” section, enter your conflict of interest statement in the “Confidential to Editor” section, and submit your "Accept" recommendation.

Reviewer #1: All comments have been addressed

Reviewer #2: (No Response)

2. Is the manuscript technically sound, and do the data support the conclusions?

Reviewer #1: Yes

Reviewer #2: Partly

3. Has the statistical analysis been performed appropriately and rigorously? 

Reviewer #1: Yes

Reviewer #2: Yes

4. Have the authors made all data underlying the findings in their manuscript fully available?

Reviewer #1: Yes

Reviewer #2: Yes

5. Is the manuscript presented in an intelligible fashion and written in standard English?

Reviewer #1: Yes

Reviewer #2: Yes

6. Review Comments to the Author

Comment reviewer 1:

Reviewer #1: Thank you to Lambooij and co-authors for the revised version of their manuscript. Following their effort, the quality and clarity of the article have improved considerably.

I have few additional minor remarks that I feel should still be addressed before publication.

Authors’ response

We thank the reviewer for the compliments and critical and constructive assessment of the paper at this current round. In the following we address the comments that were addressed point by point.

Comment reviewer 1:

General comment:

Thought the form is significantly better (with the exception of the discussion which seems of lesser quality), I would suggest an additional review to check for punctuation, spacing between words and words/punctuation, and for form (eg: Bundled Payment, with capitals or not? be consistent).

Specific comments (refer to the clean version of the manuscript, not the track/trace) (words in capital letters are my suggestions for addition):

Abstract:

-if possible (if words allow) I would contextualize the background to the Netherlands. Eg: ‘…describes a DUTCH study…’

-third line ‘….pregnant women’s….’ NOT ‘…[THE] pregnant women’s…’

Authors’ response

Thank you. We added ‘Dutch’ and deleted ‘the’ in the abstract. Furthermore we checked the consistent description of ‘Bundled Payment’ and punctuation throughout the manuscript.

Comment reviewer 1:

Introduction:

-the second sentence of the second paragraph (starting with ‘Empirical…) is not grammatically correct

Authors’ response

Thank you. We changed the sentence into the following: 

“Empirical associations have been found between use of integrated care and increased patient satisfaction, perceived quality of care and patient access to services”

-last sentence in the first page, I would specify to: ‘in THE six DUTCH regions…’

Authors’ response

We added ‘Dutch’

-second page of the introduction, second paragraph, sentence starting with ‘But also…’: is ‘birth place process’ the right term? Also, is reference #10 the correct one? I couldn’t find any info on birth place process in reference #10

Authors’ response

Thank you for your observation. It should be ‘and influence in the birth process’. We changed this accordingly. In response to this comment to find out what went wrong with the reference, we found that our literature software End Note had caused and error involving reference 10 and reference 3. They referred to the same paper but were included as different references. We corrected this and added the correct reference.

-last paragraph second page, ‘the results of this study are relevant for policymakers, as THEY PROVIDE’ and not it provides

Authors’ response

We changed this accordingly, thank you.

Comment reviewer 1:

Results:

-Sample section, last sentence: ‘…and women who were pregnant with their…  should this be: ‘…and MORE women who were pregnant with their first child….'?

Authors’ response

Indeed, adding ‘more’ adds clarity to what we mean to say. We added ‘more’ to the sentence.

Comment reviewer 1:

Discussion:

-second sentence, I would suggest: ‘For SOME women the unlimited provider choice is the most important aspect, and THEY find this more….’

Authors’ response

That is a good suggestion. Thank you. We changed this accordingly.

-first sentence second paragraph, remove the A before about: ‘…that people had [a] about a one in four….’

Authors’ response

Thank you for your thorough reading of the paper. We changed this in line with the suggestion.

-first sentence second page: ‘….difficult-to-measure (add the dash between to and measure) aspects such as [of] harder-to-measure…’ AND try not to repeat ‘such as’ in the same sentence so closely

Authors’ response

We added the dash and rephrased the sentence, removing a “such as”:

“Future studies may yield relevant new information on both relative importance and possible interactions of difficult-to-measure aspects and harder-to-affect aspects such as women’s previous experiences compared to factors that can be affected by maternity care providers or policymakers.”

-last paragraph second page, first sentence: remove the last word ‘attributes’, you already mentioned it earlier in the sentence, that is enough

Authors’ response

We removed the last word ‘attributes’.

-last line second page: scenario’s is no plural in English (that is the Dutch form), in English it’s scenarios

Authors’ response

Thank you. We corrected this.

-third page, third line: ‘intrinsic factOR’

Authors’ response

We understand the suggestion of the reviewer. However, we mean to imply that trading is a fact of life when making choices. So even though ‘factor’ would fit in this sentence. We prefer to maintain the word ‘fact’.

-end of second paragraph: ‘Since DCEs’ main objectives ARE to describe…’ OR ‘Since DCEs’ main OBJECTIVE is to describe…’

Authors’ response

Thank you. We corrected this:

“Since DCEs’ main objectives are to describe scenarios as close to a real decision making situation as possible […]”

-third paragraph: third and fourth sentence start with ‘this means’…it feels you are trying to explain a concepts with several sentences without succeeding. Can you limit the use of ‘this means’ to once?

Authors’ response

We adapted this section:

“This means that in the current study, the respondents are likely to have more information on factors that affect the quality of maternity care than in real life. Consequently the relative importance of those factors may be overestimated compared to clearly visible factors such as travel time or advice from the midwife.”

-third page last sentence: ‘….in practice THE WOMEN...’  instead of ‘they’ I would write ‘the women’, since the subject in the first part of the sentence is the study, not the women.

Authors’ response

Thank you for pointing this out. We changed ‘they’ to ‘the women’.

-fourth page, twice used: ‘medical technical measures’  is this the right term? It was not used in the text before and it’s not an obvious term

Authors’ response

We mean to refer to ‘medical technical measures of quality’. We changed this to medical technical quality measures:

“In our case, we needed to exclude more medical technical quality measures (e.g. APGAR) because our main focus was to resemble a choice similar to a real-life choice as much as possible. When it comes to quality of care, it may be useful to compare the relative importance of medical technical quality measures of care and more patient- centred measures of quality of care in future studies.”

-fourth page, third line: I would remove ‘choice similar to a’

Authors’ response

Thank you for the suggestion. We removed the words accordingly.

-the very last sentence of the discussion seems to be speculation without references. It’s a peculiar way to end a discussion. I would revise it

Authors’ response

The statement indeed may have been phrased too strongly. We rephrased the sentence to refer more clearly to the results of the current study, making clear that the authors consider this to be a policy implication of the results. 

“This study indicates that enabling provider choice under a bundled payment model is essential in order to develop delivery models that in meets the demands of all pregnant women”

Comment reviewer 1:

Conclusion:

-usually reference are not used in the conclusion, is references 14 needed there again?

Authors’ response

We agree with the reviewer and deleted the sentence. 

Comment reviewer 2:

Reviewer #2: Authors have made an effort to address my comments. However, the following comments need additional attention.

Comment reviewer 2:

• Details about the variables, including coding, and the reason for their inclusion should be presented in the methods section. Supporting file S5 Codebook to dataset is fine but it is important to provide the above information in the methods section.

Authors’ response

We thank the reviewer for critically assessing the paper and constructive comments. 

We agree with the reviewer that readers will need to be able to read more details in our methods section.

Therefore we added the following two blocks of text in the methods sections:

“This process led to the final set of attributes, levels and final rephrasing of attributes and levels (Table 1). The first three attributes are operationalisations of quality of care that can be observed by clients (indicated by a (Q) in de table), the final two attributes in table1 are operationalisations that affect the choice possibilities (C) of the women. The first attribute, “Information between care professionals by single EMR” is included to operationalize an essential part of integrating care services: access to relevant and up-to-date patient information by all care professionals [18]. Both the experts and the respondents in the pilots agreed to the relevance of this attribute. The second attribute, ‘Information to you by the midwife’ was included to operationalise the way the information from the health care organisation was given to the pregnant women. In order to make informed decisions, having access to sufficient and accurate information is essential. The third attribute ‘Advice by friends and family’ was an operationalisation of quality of care of the maternity care organisation. The Dutch National Health Care Institute (Zorginstituut Nederland) developed a set of quality indicators for maternity care. Following this set, in expert interviews and in the pilot test we tested the following options: (1) a standardized quality score on internet indicating whether the organisation was ranked in the top 20% or average, (2) percentage of complications of the maternity care organisation, presented on the internet, (3) a score on forced deliveries, presented as average, below average or above average, (4) average APGAR score of maternity care organisation, (5) reviews on the internet of experiences by other women. The experts preferred more objective measures (1-4), however, the respondents in our pilot study indicated that this information did not mean anything to them and that they would only value information related to quality of care presented to them by people that they knew. The fourth attribute ‘Organisation of maternity care’ was included to operationalize the consequences for the pregnant women of an integrated care organisation (IMCO). The final attribute ‘travel time’ was included as a consequence of integrating care; if maternity care organisations would integrate, this could imply that the nearest provider would not be cooperating with the IMCO that was chosen but women would need to travel for a longer time for their maternity care visits.”

And 

“All variables were effect coded [27], except for travel time. This was recoded as follows: 5 minutes=0,5; 10 minutes=1; 15 minutes=1,5 and 20 minutes=2. The reference category for ‘info midwife med’ and ‘info midwife max’ was ‘info midwife minimum’ (level 1 in table 1). The reference category for ‘Org. partly fixed’ and ‘Org. no choice’ was ‘all care organized separately’ (level 1 in table 1).”

Comment reviewer 2:

• Authors should extend the discussion (the first 3 paragraphs) and add an interpretation of the results based on previous literature to establish convergent validity and to outline the implications for policy and research. I mean an interpretation of the findings on women’s preferences not the methodology aspects of the study. Why are these preferences observed? How do they compare to previous studies? What do they mean for policy and research?

Authors’ response

We agree with the reviewer that this is adds relevant information to the paper. We added paragraphs to the Discussion section, comparing our study to previous studies. We focus on a number of similarities and we mention a number of new insights, adding to previous work. We also added a paragraph discussing our view on the policy implications of the findings of our study.

Paragraphs in the discussion reflecting on previous research and our results:

“Our results are in line with previous findings that women are willing to trade between factors of maternity care [30]. What this study adds to the literature is the diversification in the population of interest: different women will make different choices when they can. This is in line with the previous finding that women differ in preferences related to service attributes [31]. A difference between that review and our findings is that the current study is restricted to attributes related to organisational aspects of maternity care services that are directly influenced by the design of the bundled payment model, while the underlying studies in the review also included other aspects (e.g. personal experiences, factors related more specifically to maternity care providers involved, such as continuity of midwife) [31]. We excluded personal experiences [32, 33] women’s beliefs and values [34] or type of birth setting [35-37], even though these factors are known to be relevant and to affect the preferences of pregnant women. The downside of restricting this study to factors that can be affected by the organisation of care is that we do not know the relative importance of those factors compared to the factors we did not include. 

We also found that availability of pain relief was very important to women and partners [14, 31, 35-41]. Although pain relief is generally considered a highly important aspect of maternity care decisions, this is only very weakly linked to the organizations of integrated care and therefore not included in this study. 

For the majority of the respondents we found that the information given by the midwife was weighed to be least important, and information given by friends and family was more important. This is in line with [42] who found that midwifes were not the main source of information for pregnant women, but that women wanted the option to discuss and consider their birth preferences throughout their pregnancy with multiple sources, not at a fixed point. This would mean that policymakers nor health care professionals are able to provide pregnant women with the type of information that they need in order to develop their preferences.”

Paragraph on policy implications:

“Current design of the Dutch bundled payment guarantees an unlimited choice for pregnant women via the existence of the bundle breaker. In practice, this means that when a pregnant women receives care from a care provider, which is not subcontracted by the IMCO, the bundled payment expires and all maternity care services are reimbursed via the existing payment model which is predominantly fee-for-services. This leads to an enormous administrative burden for care providers. In order to keep support among care providers for the bundled payment model the design of the bundle breaker preferably must be simplified in order to find a middle ground between unlimited provider choice and the administrative burden for care providers. Finding this optimum is not straight forward but is needed to keep support among both pregnant women and providers. The results of this study support current policy design of the bundle breaker, indicating that the current of the bundled payment with a bundle breaker appears to be close to the optimum of preferences of pregnant women.”

Comment reviewer 2:

• The selection bias and the limitations of it should be discussed more substantially to establish the possibility to extrapolate the results. Unless this is done, conclusions only refer to women in the register “The happy box” and not to all women in the Netherlands. The representativeness of the sample should be accounted for when making conclusions. In this regard, the validity of the current conclusions is unclear.

Authors’ response

The reviewer appears to consider the possibility that selection bias in the sample may be caused by the source we used to approach our respondents, affecting the results. We also considered this risk in advance and we therefore included control variables in the questionnaire to correct for possible biases. We selected the variables based on previous work and more or less ad hoc hypothesis of aspects that might affect preferences of pregnant women. We tested whether these sample characteristics affect the results, this turned out to be true for educational level, therefore we included educational level in the model. We consider the channel through which we contacted the hard-to-reach respondent group as instrumental, and tried to control for risks of bias in the analyses. 

In the paper, the most important conclusions are drawn considering existing preference structures in the population. We presented our analyses to determine relevant class predictors “We tested the following parameters for a significant contribution to the class assignment model: education high/low, urbanisation level (5 categories), urbanisation city/rural, first child/gravidity status. ” 

We found that educational level significantly affects preference structures, and the other aspects do not. We therefore included educational level as class predictor. We referred to evidence that higher educated respondents are likely to trade a larger number of attributes in their choices. We have no empirical indications nor hypotheses to predict directions of differences between people as to why having a first child or second/third/more, would affect the preferences of the women. 

We therefore have no reason to assume that the known discrepancies between our sample and the general population would affect the four preference structures that we found. A different sample would likely give slightly different parameters, which may affect the relative importance of attributes within preference structures. However, given the sizes of most RIS’s it we find it unlikely that a another sample would yield shifts that are sufficiently large to thoroughly affect the ranking of the attributes.

Based on this, we have no reason to assume that in a sample with more lower educated women, more women who live in cities and more women with a multigravida status, different preference structures are likely to emerge.

Conclusions that we draw on the prevalence of the preference structures are based on the chance that each respondent has to belong to a class. Also here, a different sample is likely to yield slightly different parameters. However in the current sample the odds are divided evenly and the conclusions we draw take account of the uncertainty. Even though we cannot exclude that a different sample would yield different results, we see no reason why a different sample would lead to completely different results and affect the main conclusions to a large extent. We added these arguments to the text. 

The section on the sample is now as follows:

“We found that in our sample, highly educated women, women from rural areas and women who were expecting their first child were overrepresented. Previous research has shown that higher educated respondents trade more of the included attributes than lower educated respondents in DCEs [42]. This could mean that in real life choices, a larger proportion of women weigh fewer of the maternity care aspects than the aspects that were included in the current study. Despite the overrepresentation of highly educated respondents, the sample benefits from the fact that all respondents had actual experience in the choice in the DCE because they were pregnant at the time of filling in the questionnaire. This meant that respondents were familiar with and had experience in the decisions they were asked to make in DCEs. In the analyses, we found that level of urbanisation, gravidity status did not predict the classes in the LCA. This implies that these person related factors do not affect the preferences structures that we found. If we would have had a sample with more highly educated women, living in a city it is likely that the parameters we would have found would differ. However, considering the relative importance of the attributes in the four preference structures, these differences in parameters would need to be very large to affect the ranking, and therefore the conclusions that we draw on the existence of the preference structures. We have therefore no reason to assume that the relative importance of the attributes in our results differ from the relative importance in real life choices, when just looking at the ranking of attributes. However the precise parameter values are likely to differ between the sample in this study and the general population [43]. We have therefore no reason to assume that a different sample would lead to very different conclusions concerning the preference structures of pregnant women in the Netherlands. ”

7. PLOS authors have the option to publish the peer review history of their article (what does this mean?). If published, this will include your full peer review and any attached files.

Do you want your identity to be public for this peer review? For information about this choice, including consent withdrawal, please see our Privacy Policy.

Reviewer #1: Yes: Elena Ambrosino

Reviewer #2: No

---

## [Decision Letter · Decision Letter 2]

8 Apr 2020

Trading patients’ choice in providers for quality of maternity care? A discrete choice experiment amongst pregnant women

PONE-D-19-23167R2

Dear Dr. Lambooij,

We are pleased to inform you that your manuscript has been judged scientifically suitable for publication and will be formally accepted for publication once it complies with all outstanding technical requirements.

With kind regards,

Federica Angeli

Academic Editor

PLOS ONE

Additional Editor Comments (optional):

Reviewers' comments:

Reviewer's Responses to Questions

**Comments to the Author**

1. If the authors have adequately addressed your comments raised in a previous round of review and you feel that this manuscript is now acceptable for publication, you may indicate that here to bypass the “Comments to the Author” section, enter your conflict of interest statement in the “Confidential to Editor” section, and submit your "Accept" recommendation.

Reviewer #2: All comments have been addressed

2. Is the manuscript technically sound, and do the data support the conclusions?

Reviewer #2: Yes

3. Has the statistical analysis been performed appropriately and rigorously? 

Reviewer #2: Yes

4. Have the authors made all data underlying the findings in their manuscript fully available?

Reviewer #2: Yes

5. Is the manuscript presented in an intelligible fashion and written in standard English?

Reviewer #2: Yes

6. Review Comments to the Author

Reviewer #2: All my comments have been addressed. This is a relevant paper addressing an important healthcare topic.

7. PLOS authors have the option to publish the peer review history of their article (what does this mean?). If published, this will include your full peer review and any attached files.

Reviewer #2: No

---

## [Editor Report · Acceptance letter]

13 Apr 2020

PONE-D-19-23167R2 

Trading patients’ choice in providers for quality of maternity care? A discrete choice experiment amongst pregnant women 

Dear Dr. Lambooij:

I am pleased to inform you that your manuscript has been deemed suitable for publication in PLOS ONE. Congratulations! Your manuscript is now with our production department. 

With kind regards,

on behalf of

Prof. Federica Angeli 

Academic Editor

PLOS ONE